# Invasion of African *Clarias gariepinus* Drives Genetic Erosion of the Indigenous *C. batrachus* in Bangladesh

**DOI:** 10.3390/biology11020252

**Published:** 2022-02-06

**Authors:** Imran Parvez, Rukaya Akter Rumi, Purnima Rani Ray, Mohammad Mahbubul Hassan, Shirin Sultana, Rubaiya Pervin, Suvit Suwanno, Siriporn Pradit

**Affiliations:** 1Faculty of Environmental Management, Prince of Songkla University, Hat Yai 90110, Songkhla, Thailand; iparvez.fbg@hstu.ac.bd (I.P.); suvit.su@psu.ac.th (S.S.); 2Department of Fisheries Biology and Genetics, Hajee Mohammad Danesh Science and Technology University, Dinajpur 5200, Bangladesh; rukaya.1605523@std.hstu.ac.bd (R.A.R.); purnima.1805440@std.hstu.ac.bd (P.R.R.); hassanm@ufl.edu (M.M.H.); 3School of Forest, Fisheries, and Geomatic Sciences, University of Florida/IFAS, 7922 NW 71st Street, Gainesville, FL 32653, USA; 4Fisheries Biotechnology Division, National Institute of Biotechnology, Ministry of Science and Technology, Dhaka 1000, Bangladesh; shirin@nib.gov.bd; 5Department of Fisheries Management, Hajee Mohammad Danesh Science and Technology University, Dinajpur 5200, Bangladesh; rubaiya2015@hstu.ac.bd; 6Coastal and Climate Change Research Center, Faculty of Environmental Management, Prince of Songkla University, Hat Yai 90110, Songkhla, Thailand

**Keywords:** catfish, *Claris batrachus*, *C. gariepinus*, aquaculture, Bangladesh

## Abstract

**Simple Summary:**

Bangladesh has substantially increased aquaculture production over the last few decades, and the exotic species share a significant portion of the total fish production. Although exotic species are contributing to aquaculture production, a few of them are causing biodiversity loss and genetic erosion of native species. The African catfish *Clarias gariepinus* is a highly carnivorous species and predates small indigenous freshwater fishes when escaping into natural water bodies. In addition, the hybridization of *C. batrachus* and *C. gariepinus* is considered a threat to the indigenous population. Although the government of Bangladesh has banned the farming of *C. gariepinus*, this species has been identified in local markets, and evidence of hybridization between *C. gariepinus* and *C. batrachus* has been found. This study revealed genetic erosion of native *C. batrachus* by the gene sequences of cytochrome c oxidase subunit I and cytochrome b. The phylogenetic tree confirmed the occurrences of hybridization between *C. gariepinus* and *C. batrachus.* Genetic erosion in the native catfish population is alarming for aquaculture sustainability and biodiversity conservation in Bangladesh.

**Abstract:**

The African catfish *Clarias gariepinus* has been introduced for aquaculture in Bangladesh due to the scarcity of indigenous *C. batrachus* fingerlings. However, the government of Bangladesh has banned the farming of *C. gariepinus* due to the carnivorous nature of this species. Recently *C. gariepinus* has been reported by fish farmers and consumers in Bangladesh, and unplanned hybridization between native and exotic species has been suspected. This study attempts to know the purity of *C. batrachus* by analyzing mitochondrial genes. Both directly sequenced and retrieved Cytochrome C Oxidase subunit I (COI) and cytochrome b (Cytb) genes from *C. gareipinus* and *C. batrachus* were analyzed by MEGA software. The morphologically dissimilar *C. batrachus* showed the least genetic distance (0.295) from *C. gariepinus*, which provided evidence of hybridization between the two species. Maximum likelihood (ML) phylogenetic trees showed that *C. batrachus* from Bangladesh did not cluster with *C. batrachus* of other countries, instead *C. batrachus* clustered with the exotic *C. gariepinus*. The suspected hybrid formed sister taxa with the exotic *C. gariepinus*. The study corroborates the genetic deterioration of *C. batrachus* by unplanned hybridization with the invasive *C. gariepinus*. Unplanned hybridization has deleterious consequences; therefore, immediate action is necessary for aquaculture sustainability and biodiversity conservation in Bangladesh.

## 1. Introduction

The walking catfish *Clarias batrachus* belonging to the Clariidae family, commonly known as walking catfish, is a popular food fish in the Indian sub-continent, including Bangladesh [1,2,3]. Catfish are a diverse group of fishes distributed from freshwater to marine environments, such as *Clarias batrachus* in freshwater [4] and *Arius maculatus* in brackish water [5]. Three species of the *Clarias* genera are important aquaculture candidates in Asia, namely *C. batrachus* in the Indian subcontinent [6], *C. fuscus* in Taiwan region [7,8], and *C. macrocephalus* in South-East Asia [9]. The native range of *C. batrachus* is Asia but *Clarias* aff. *Batrachus* from Indochina and Sundaland has been incorrectly identified as *Clarias batrachus* from Java [10]. The widely distributed African catfish *C. gariepinus* expanded its native range from the South to the Middle East and Eastern Europe [11].

Many exotic species have been introduced in Bangladesh for aquaculture in the past few decades, and some species are playing important roles in aquaculture production. A few of the exotic species have been proven detrimental to aquatic biodiversity and eventually banned for aquaculture in Bangladesh. The African magur, *Clarias gariepinus*, was introduced in Bangladesh in 1989 from Thailand by the Ministry of Fisheries and livestock (MoFL), but banned from aquaculture since 2014. The introduction of *C. gariepinus* caused native biodiversity loss due to its predatory nature [12]. The feasibility of hybrid vigor production between *C. batrachus* × *C. gariepinus* has been assessed but attained limited success [13]. Hybridization of *C.*
*gariepinus* male × *C. macrocephalus* female in Vietnam has been widely practiced [14]. However, the potential of hybrids for a decline in the abundance of *C. batrachus* in the Chao Phraya and Mekong Basins has been considered [13]. Although aquaculture of *C. gariepinus* has been banned in Bangladesh [15], the availability of this species has been reported by consumers and farmers. Hybridization of *C. batrachus* × *C. gariepinus* has been known to occur in India [16] and in the Mekong Basin of Vietnam and Thailand [13]. Unplanned hybridization between *C. gariepinus* and *C. batrachus* is considered illegal in Bangladesh. Hybrids are less accepted by consumers due to differences in appearance and taste. In addition, the escape of the hybrids in natural water bodies could cause irreversible loss of the native biodiversity. If the purity of native stock is compromised in the natural population, it would be an irreversible problem for the future.

The mitochondrial genes were analyzed in this study to identify the native and hybrid *Clarias* species in Bangladesh. DNA sequencing was used to identify animal species because the mutation rate is low enough to allow distinguishing of closely related species [17]. This approach is used in evolutionary biology, phylogenetic systematics, and population genetics. Mitochondrial genes based on cytochrome C oxidase subunit I (COI) and cytochrome b (cytb) have been targeted for analysis in this study. These mitochondrial genes have been used in evolutionary and phylogenetic analyses of fishes, including genetic variation assessment of native and exotic climbing perch *Anabas testudineus* in Bangladesh [18]. The COI nucleotide sequences are widely used to authenticate the purebreds of different fish species and families, including the Sparidae family [19], groper fishes [20], Cyprinidae family [21], *Ompok* genus [22], and Clariidae and Pangasiidae families [23]. The mitochondrial cytochrome b (Cytb) has been found effective in fish species identification and authentication [24,25], the identification of catfishes in Korea [26], *Puntius* genus in Indian rivers [27], and fishes of the South China Sea [25]. In this study, COI and Cytb based DNA barcoding techniques have been targeted to identify *C. batrachus*, *C. gareipinus*, and the suspected hybrids in Bangladesh.

## 2. Materials and Methods

Fish collection and morphological identification: A total of 120 specimens of *C. batrachus* and 20 specimens of *C. gariepinus* were collected from three wholesale fish markets in Dinajpur district, Bangladesh. These fishes were produced by semi-intensive culture in earthen ponds. Traders in these fish markets collect fishes from all over the country, thereby the specimen used in this study were sourced from different parts of Bangladesh. Out of 120 specimens, a total of 20 native and 20 suspected hybrids were identified based on distinguishable morphological differences in body color, pectoral spine, and occipital process [13,28]. The body colors were brownish-black and greyish-black for native *C. batrachus* and the suspected hybrid, respectively (Figure 1). The occipital process was narrow and pointed in native *C. batrachus*, broad and blunt in suspected hybrid. Pectoral fin was elongated with strong spine in native *C. batrachus* (1), whereas the pectoral fin was round-shaped, without spines and rayed in suspected hybrid (Figure 1). The exotic *C. gariepinus* used were clearly distinguishable from the native *C. batrachus* (1) and suspected hybrid; therefore, morphological traits were not described in this study. Geometric morphometry was then used to validate the accuracy of morphological identification among the native *C. batrachus*, suspected hybrid and exotic *C. gariepinus*. A total of 13 landmark points from each fish specimen were digitized and analyzed by TPS software series and Past software (Ver. 3.0).

Tissue sample collection and DNA extraction: Fish muscle samples were collected and preserved in 95% ethanol. Phenol-Chloroform-Isoamyl alcohol method was used to extract DNA from the preserved muscle tissue, according to Shafi et al. 2016 [29]. The extracted DNA was confirmed by visualization of bands at 1% agarose gel electrophoresis with ethidium bromide (5 µg/mL) incorporated in 1× TBE buffer. Finally, DNA concentration was quantified by optical density (OD) measurement using a spectrophotometer at 260 nm (Thermo Scientific™, Waltham, MA, USA, NanoDrop™ 2000: Model SIA340). The step-by-step process from sample collection to species identification is presented in Figure 2.

Amplification mitochondrial COI and Cytb genes: The COI gene sequences (648 bp) were amplified using the primer pair Fish-F1 (5′-TCA ACC AAC CAC AAA GAC ATT GGC AC-3′), Fish-R1 (5′-TAG ACT TCT GGG TGG CCA AAG AAT CA-3′). A total of 25 µL reaction mixture was used that contained 1× assay buffer (100 mM Tris, 500 mM KCL, 0.1% gelatin, at pH 9.0), 1.5 mM MgCl_2_, 5 pmol of each primer, 200 µM of each dNTP, 1.5 U Taq DNA polymerase, and 20 ng of template DNA. The following protocol was used for PCR amplification: 95 °C for 2 min, followed by 35 cycles of 94 °C (30 s), 54 °C (30 s), and 72 °C (1 min), and extension for 2 min at 72 °C (Figure 2c). Partial fragments of 360 bp mitochondrial Cytb gene were amplified using primers CytbF (5′-AAA AAG CTT CCA TCC AAC ATC TC-3′) and CytbR (5′-AAA CTG CAG CCC CTC AGA ATG AT-3′). Again, a total of 25 µL reaction mixture was used that contained 12.5 µL master mix (Taq DNA Polymerase, dNTPs, MgCl_2_), 5 pmol of each primer, 20 ng of template DNA, and 9.5 µL distilled water. PCR amplification protocol included: 104 °C for 2 min, 95 °C for 1 min followed by 35 cycles of 94 °C (1 min), 48 °C (1 min), 72 °C (45 s), and extension for 5 min at 72 °C.

Purification of PCR products and sequencing: The amplified PCR products were passed through 1.5% agarose gels, stained with ethidium bromide, and visualized under UV illumination in the Gel-Doc system (BIO-RAD, Hercules, CA, USA). The PCR products were purified according to the manufacturer’s instructions of the Invitrogen purification kit (K310001, Thermo scientific, Waltham, MA, USA). The purified products were labeled by the Big Dye Terminator V.3.1 cycle sequencing kit (Applied Biosystems Inc, Thermo scientific, Waltham, MA, USA) and were sequenced bidirectionally using ABI 3730 capillary sequencer (Figure 2d).

**Sequence analysis:** BioEdit sequence alignment editor (version 7.0.5.2) was used to edit and align the raw DNA sequences (Figure 2e) [30]. The COI and Cytb sequences were aligned, and the final consensus size was identified. The consensus size of the COI and Cytb genes was 646 bp to 682 bp, and 356 bp to 368 bp, respectively. The Basic Local Alignment Search Tool (BLAST) at the NCBI website (http://ncbi.nih.gov/BLAST/ accessed on 11 October 2021) was used to process DNA sequences. Then the sequences were submitted to the NCBI GenBank database via the BankIt submission tool. Nucleotide sequences of the COI and Cytb genes from *C. batrachus* and *C. gariepinus* from other countries were retrieved from the NCBI GenBank database (http://www.ncbi.nlm.nih.gov/ accessed on 11 October 2021) to include reference sequences (Table 1). The polymorphic sites, nucleotide diversity, nucleotide composition, the disparity between sequences, net base composition bias, transitional/transversional bias, a test of homogeneity of substitution patterns between sequences, sequence divergence within species based on the Kimura 2 Parameter (K2P) model were run in MEGA (Version 7.01) for molecular genetic analysis [31]. The larger differences for all sequences were estimated by the Monte Carlo test (500 replicates) to observe the differences of base composition biases from the expected bases [31]. Based on the extent of differences in base composition biases between sequences, the probability of rejecting a null hypothesis that the sequences have evolved with the same pattern of substitution was tested [32]. The transition/transversion biases of COI gene sequence substitutions were estimated using a Kimura 2-parameter (K2P) model [33]. Phylogenetic trees based on a maximum likelihood method using COI and Cytb gene sequences were constructed using MEGA, which was validated using 1000 replicates of bootstrap values. Based on the results, *C. batrachus*, *C. gariepinus*, and the suspected hybrids were identified.

## 3. Results

### 3.1. Sequence Analysis of COI and Cytb Genes

Validation of morphological identification by geometric morphometry: Based on geometric morphometry, the scattered plot of the landmark points showed that native *C. batrachus*, suspected hybrid and exotic *C. gariepinus* were morphologically distinguishable (Figure 3). The principal component 1 (PCA 1)_and PCA 2 of landmark points showed that native *C. batrachus*, suspected hybrid exotic *C. gariepinus* formed a separate cluster.

### 3.2. Sequence Analysis of COI and Cytb Genes

In the COI gene, 751 sites were identified where 76 (10.12%) were conserved; 670 (89.88%) were variable, which indicated a high level of variation among *C. batrachus*, *C. gariepinus*, and the hybrid. The average nucleotide composition of the COI gene was 29.3%, 26.6%, 24.6%, and 19.5% for T, C, A, and G, respectively. The AT content was higher than GC content in all the samples. Higher differences were observed between the sequence pair of *C. batrachus* of Bangladesh and reference *C. batrachus* from the Philippines and Indonesia, Sequenced *C. gariepinus*, and reference *C. batrachus* from the Philiphines and Indonesia. Higher differences were not observed between the sequenced *C. gariepinus* and suspected hybrid (Table 2).

### 3.3. Transition/Transversion Bias

The estimated transition/transversion bias (*R*) among the COI gene sequences of the *Clarias* genus was 1.12. The rates of transitional and transversional substitutions are presented in bold and italics, respectively (Table 3). The nucleotide frequencies were A = 24.91%, T/U = 29.34%, C = 26.39% and G = 19.36%. The rates of transitional substitution from A to G, T to C, C to T, and G to A were 8.76%, 15.39%, 17.11%, and 11.27%, respectively (Table 2). The rates of transversional substitution from A to T, A to C, T to A, T to G, C to A, C to G, G to T, and G to C were 6.96%, 6.26%, 5.9107%, 4.59%, 6.96%, and 6.26%, respectively (Table 3). The rates of transitional substitution of the COI nucleotide sequences were higher (52.547%) than transversional substitution (47.453%).

### 3.4. Homogeneity of Substitution Patterns of COI Sequences

The estimated *p* values smaller than 0.05 (considered significant and marked with an asterisk) are presented above the diagonal in Table 4. The estimated disparity index per site is presented for each sequence pair above the diagonal in Table 4. The *p*-values were larger than 0.05 between sequence pairs of reference *C. batrachus* from the Philippines and Indonesia; the reference *C. batrachus* from India and Bangladesh relative to the reference from the Philippines and Indonesia; the Bangladeshi sequence of native *C. batrachus* and *C. gariepinus*; the *C. gariepinus* from Nigeria, Indonesia and *C. batrachus* from India, Indonesia, and the Philippines; the sequence of suspected hybrid and native *C. batrachus* and exotic *C. gariepinus*. The sequences of the COI gene in native *C. batrachus* evolved with the same pattern of substitution like the species originating from other counties (Table 4).

### 3.5. Genetic Distance

The lowest genetic distance was found between suspected hybrid and *C. gariepinus* (0.295). The highest genetic distance (6.238) was found between *C. gariepinus* from Indonesia and Bangladeshi *C. batrachus*. Low levels of genetic divergence (0.295–0.339) were found among sequenced native *Clarias batrachus*, suspected hybrid and exotic *C. gariepinus* (in Bangladesh, whereas interspecies genetic divergence was found between *C. batrachus* and *C. gariepinus* from other countries (Table 5). The pairwise genetic distances based on nucleotide sequence divergences are presented in Table 5.

### 3.6. Phylogenetic Tree Using COI Nucleotide Sequences by Maximum Likelihood Methods

The phylogenetic tree inferred from COI nucleotide sequences based on the maximum likelihood method showed that all the sequenced native *C. batrachus* and exotic *C. gariepinus* formed a single clade, where *C. gariepinus* and suspected hybrid and *C. gariepinus* formed a sister clade. The sequenced *C. batrachus* of Bangladeshi did not cluster with any of the reference sequences *C. batrachus* originated from other countries. However, *C. batrachus* from India, Indonesia, and the Philippines formed one clade. *C. gariepinus* from Nigeria and Indonesia formed a sister clade (Figure 4).

### 3.7. Phylogenetic Tree Using Cytb Nucleotide Sequences by Maximum Likelihood Method

The phylogenetic tree based on the Cytb nucleotide sequences was used to validate the phylogenetic tree inferred from the COI nucleotide sequence (Figure 5). The results showed that sequenced suspected hybrid formed sister taxa with the sequenced *C. gariepinus*. The native *C. batrachus* formed a clade with *C. batrachus* of originated from different countries (Nigeria, France, Germany, and Malaysia) except suspected hybrid and *C. gariepinus*. The formation of sister clade of suspected hybrid with native *C. batrachus* and *C. gariepinus* confirmed the occurrence of hybridization of *C. batrachus* and *C. gariepinus* in Bangladesh, which validated the results based on COI gene.

## 4. Discussion

In the present study, six partial sequences (forward and reverse) from native *C. batrachus*, *C. gariepinus*, and suspected hybrid were 751, 746, and 728, respectively. In multiple sequence alignment, a total of 751 sites were identified: 76 (10.11%) of them were conserved, 670 (89.21%) were variable, 670 (89.21%) were informative for parsimony sites, 653 (86.95%) were singleton sites. In *Clarias gariepinus*, *Coptodon zillii*, and *Sarotherodon melanotheron* from Southwestern Nigeria, 49.86% were conserved sites, 49.71% were variable sites, 32.33% were parsimony informative sites, and (17.39%) were singleton sites [34]. *C. gariepinus*, *C. magur*, and *C. dussumieri* based on COI gene sequences, showed a consensus-sized length of 623 sites, where 425, 197, 163, and 34 were conserved, variable, parsimony informative, and singletons, respectively [35]. Out of 628 sites in gene sequences in the indigenous barp *Pethia manipurensis* 144, 459, 267, and 192 were conserved, variable, parsimony informative, and singletons, respectively [36].

The average nucleotide composition of COI gene in three *Clarias* populations and reference population were 29.3%, 26.6%, 24.6%, and 19.5% for T, C, A, and G, respectively. In this study, GC contents of native *Clarias batrachus* (MG988399), *C. gariepinus* (MG988400), and suspected hybrid (MG988401) were 47.1%, 49.2%, and 51%, whereas AT contents were 52.8%, 50.8%, and 48.9%, respectively. The AT content was higher than GC content in the native and exotic species and the suspected hybrid. Since AT content positively relates to the rate of evolution, a high level of mutation rate has been expected based on the results in this study. COI genes from *C. gariepinus*, *C. zillii*, and *S. melanotheron* were higher than 500 bp in which the nucleotide composition were 29.0% for T, 26.6% for C, 26.4% for A, and 18.0% for G [34]. The highest and the lowest GC contents were found in *C. zilli* (49.6%) and *C. gariepinus* (42.2%), respectively [34].

Based on the substitution pattern, the highest interspecific divergence was 6.238 between *C. gariepinus* (Indonesia) and *C. batrachus* (India), but the lowest intraspecific divergence was 0.295 between suspected hybrid (MG988401) and *C. gariepinus* (MG98400). Multiple groups from the same common ancestor evolved and accumulated differences through evolutionary divergence, which resulted in differentiation in body structure and formed a new species. The lowest divergence was found between suspected hybrid (MG988401) and *C. gariepinus* (MG98400); these were evolved and accumulated differences. Genetic distances among *C. gariepinus*, *C. zillii*, and *S. melanotheron* from southwestern Nigeria showed the highest nucleotide divergence of COI gene in *C. zilli (*π = 0.184), while *S. melanotheron* showed the lowest divergence (π = 0.065) [35]. The interspecies genetic divergence ranged from 0.056 to 0.182 for the COI gene. The mean genetic difference between *C. dussumieri* and other African catfish species was 12.1% based on COI gene sequences. The level of substitution of *C. batracus* by *C. gariepinus* in India showed that the intraspecific divergence (0.53) and intrageneric divergence (15.05) was sufficiently high to allow differentiation of these species based on DNA barcode analysis with cytochrome c oxidase subunit I (COI) gene [15].

At low levels of divergence, transitions occur more commonly than transversions in the mtDNA. In this study, the transitions versus transversion bias were 1.12 for COI. The ratio of si/sv^®^ with COI gene sequences was 0.79 for *C. gariepinus*, *C. zillii*, and *S. melanotheron* from southwestern Nigeria [34]. The indigenous and exotic *Anabas testudineus* showed a transition transversion bias of 0.94 [21].

In both phylogenetic trees, a sister group formed between the *C. gariepinus* (Nigeria) and *C. gariepinus* (Indonesia), sequenced *C. gariepinus* and sequenced suspected hybrid, whereas formed sub-clade with the native *C. batrachus*. This indicates occurrence hybridization between *native C. batrachus* with the exotic African catfish *C. gariepinus*. The feasibility of hybridization between *C. batrachus* × *C. gariepinus* was previously discussed in Bangladesh [15]. Hybridization between *C.*
*gariepinus* male × *C. macrocephalus* female is widely practiced in Vietnam [16]. The abundance of crossbreeds of the native *C. batrachus* and exotic *C. gariepinus* in Bangladesh could deteriorate the purity of the native species. Hybridization caused a decline in the abundance of *C. batrachus* in the Chao Phraya and Mekong basins [16]. Thirty species of the family Carangidae were studied using cytochrome c oxidase I (COI), in which all the described species formed monophyletic clusters in phylogenetic trees [37]. ML tree from the COI nucleotide sequences of *C. gariepinus*, *C. zillii*, and *S. melanotheron* showed cluster formation independently within their corresponding genera [34]. Similar findings were observed in catfishes, *Ompok pabda, O. pabo*, and *O. bimaculatus* from Indian rivers, which resolved phylogenetic ambiguity using the COI gene [22]. The phylogenetic relationships of native and introduced catfishes in the Philippines showed sharing of the same unique concatenated sequence by *Arius manillensis* and *A. dispar*, but two specimens of *Pterygoplichthys pardalis* and one specimen of *P. disjunctivus* shared a unique concatenated sequence with [38].

## 5. Conclusions

This study confirms that the indigenous *C. batrachus* of Bangladesh is distinctly separate from the *C. batrachus* of other countries. In this study, twenty specimens the native and the suspected hybrid catfish were used for sequence data analysis which is considered reliable for molecular phylogeny. A total of 3 to 36 specimens were used in previous studies for COI and Cytb gene biomarkers for phylogenetic analysis of different catfish species [22,23,26]. The native *C. batrachus* of Bangladesh formed a cluster with both the introduced exotic African *C. gariepinus* and the suspected hybrid. The level of genetic variation and cluster formation of the hybrid with the invasive *C. gariepinus* is alarming for the native species in Bangladesh. We call for an immediate intervention to prevent unplanned hybridization between native and exotic species. Upon the escape of the hybrids into the wild due to flood or a human mistake, the loss of purity in the wild population will be irreversible.

## Figures and Tables

**Figure 1 biology-11-00252-f001:**
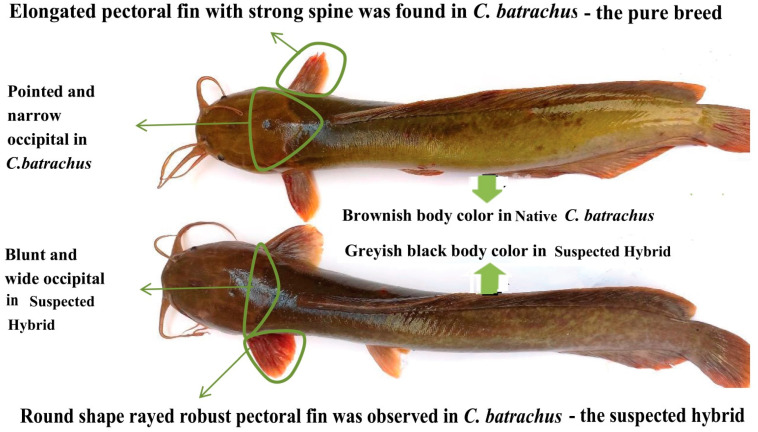
Identification of native *Clarias batrachus* and suspected hybrid based on morphological characters.

**Figure 2 biology-11-00252-f002:**
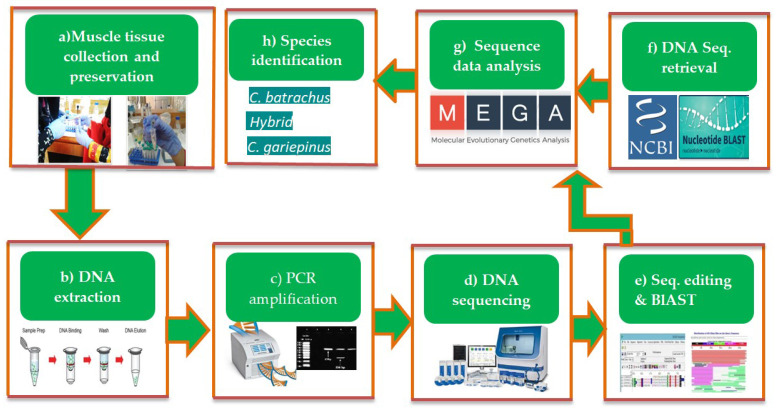
The flow chart of the experimental procedures from sample collection to DNA sequence analyses for species identification.

**Figure 3 biology-11-00252-f003:**
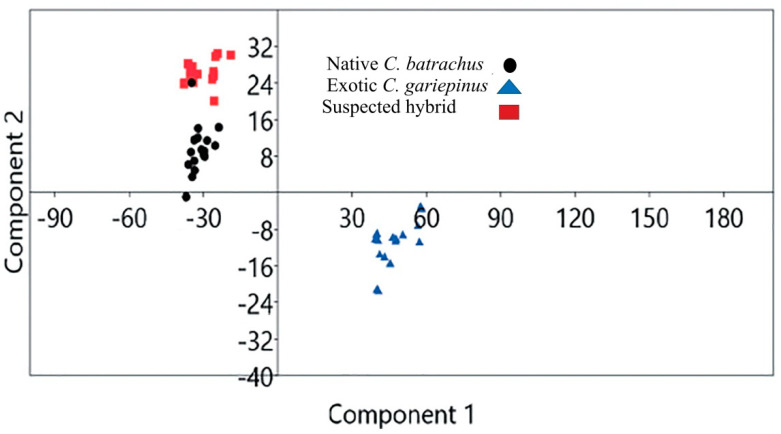
Scatter plot generated 13 landmark landmark points based on geometric morphometry of native *Clarias batrachus*, suspected hybrid and exotic *C. gariepinus*.

**Figure 4 biology-11-00252-f004:**
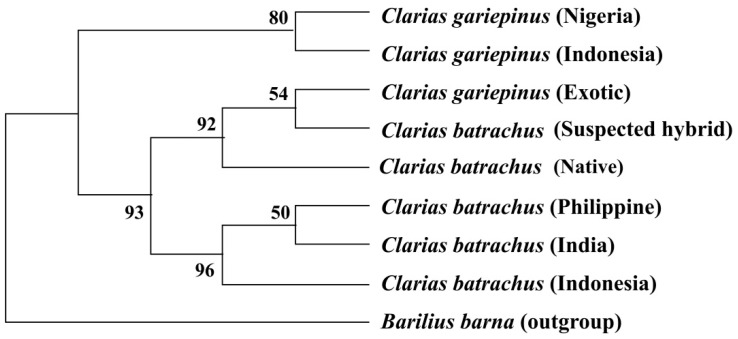
Rooted phylogenetic tree inferred from COI nucleotide sequences using maximum likelihood method. The numbers indicate the bootstrap value, which validates the sister taxa and clade formation in the phylogenetic tree.

**Figure 5 biology-11-00252-f005:**
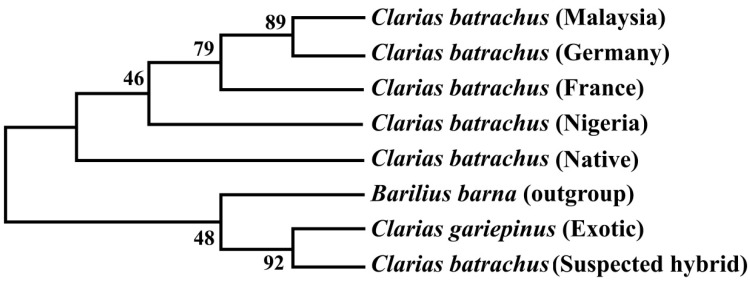
Unrooted phylogenetic tree inferred from Cytb nucleotide sequences using a maximum likelihood method. The numbers indicate the bootstrap value, which validates the sister taxa and clade formation in the phylogenetic tree.

**Table 1 biology-11-00252-t001:** The sequenced and reference COI and Cytb sequences of native *Clarias batrachus*, suspected hybrid and *C. gariepinus*.

Gene Name	Accession No	Species Name	Status	Country
COI	MG988399	*C. batrachus*	Sequenced	Bangladesh
	MG988400	*C. gariepinus*	Sequenced	Bangladesh
	MG988401	Suspected hybrid	Sequenced	Bangladesh
	HQ682681.1	*C. batrachus*	Reference	Philippine
	KU692438.1	*C. batrachus*	Reference	Indonesia
	KF214296	*C. batrachus*	Reference	India
	KU69244.1	*C. gariepinus*	Reference	Indonesia
	HM882828.1	*C. gariepinus*	Reference	Nigeria
Cytochrome b	MG988402	*C. batrachus*	Sequenced	Bangladesh
	MG988403	*C. gariepinus*	Sequenced	Bangladesh
	MG988404	Suspected hybrid	Sequenced	Bangladesh
	KR007705.1	*C. batrachus*	Reference	Germany
	JF280859.1	*C. batrachus*	Reference	Malaysia
	GU906881.1	*C. gariepinus*	Reference	Nigeria
	AF235924.1	*C. gariepinus*	Reference	France

**Table 2 biology-11-00252-t002:** The estimated net base composition bias disparity between COI nucleotide sequences of native *Clarias batrachus* suspected hybrid and exotic *C. gariepinus*.

No	Species Name	1	2	3	4	5	6	7
1	*C. batrachus* 1 (MG988399)							
2	*C. batrachus* (Philiphine)	0.189						
3	*C. batrachus* (Indonesia)	0.691	0.000					
4	*C. batrachus* (India)	0.000	0.000	0.000				
5	*C. gariepinus* (MG988400)	0.000	1.328	1.961	0.900			
6	*C. gariepinus* (Indonesia)	0.318	0.000	0.000	0.000	1.370		
7	*C. gariepinus* (Nigeria)	0.146	0.000	0.000	0.000	1.068	0.000	
8	Suspected hybrid (MG988401)	0.000	1.258	1.909	0.923	0.000	1.440	1.235

The values (>0) in the table indicate the larger differences in base composition biases than the expected bases between the studied COI nucleotide sequences [31].

**Table 3 biology-11-00252-t003:** The estimated maximum likelihood pattern of nucleotide substitutions.

	A	T	C	G
**A**	-	*6.9605%*	*6.2622%*	**8.7613%**
**T**	*5.9107%*	-	**15.3969%**	*4.5926%*
**C**	*5.9107%*	**17.1140%**	-	*4.5926%*
**G**	**11.2759%**	*6.9605%*	*6.2622%*	-

The rates of different transitional substitutions are presented in bold, and transversions substitutions are presented in italics.

**Table 4 biology-11-00252-t004:** Test of the homogeneity of substitution patterns between COI sequences of native *Clarias batrachus*, suspected hybrid and exotic *C. gariepinus*.

Sl.No	Species Name	1	2	3	4	5	6	7	8
1	*C. batrachus* 1 (MG988399)		0.189	0.691	0.000 *	0.000 *	0.318	0.146	0.000 *
2	*C. batrachus* (Philiphine)	0.194		0.000 *	0.000 *	1.328	0.000 *	0.000 *	1.258
3	*C. batrachus* (Indonesia)	0.108	1.000		0.000 *	1.961	0.000 *	0.000 *	1.909
4	*C. batrachus* (India)	1.000	1.000	1.000		0.900	0.000 *	0.000 *	0.923
5	*C. gariepinus* (MG988400)	1.000	0.044	0.012	0.100		1.370	1.068	0.000 *
6	*C. gariepinus* (Indonesia)	0.234	1.000	1.000	1.000	0.032		0.000 *	1.440
7	*C. gariepinus* (Nigeria)	0.332	1.000	1.000	1.000	0.050	1.000		1.235
8	Suspected hybrid (MG988401)	1.000	0.024	0.022	0.082	1.000	0.026	0.046	

Note: * means the *p* values smaller than 0.05 (*p* < 0.05) are considered as significant, which indi-cating the rejection of the null hypothesis that the sequences have evolved with the same pattern of substitution.

**Table 5 biology-11-00252-t005:** Pairwise genetic distances of *Clarias* species from different countries based on a Kimura-2 parameter (KP2) model.

No	Species Name	1	2	3	4	5	6	7	8
1	*C. batrachus* 1 (MG988399)								
2	*C. batrachus* (Philiphine)	4.931							
3	*C. batrachus* (Indonesia)	3.283	3.110						
4	*C. batrachus* (India)	4.122	2.137	2.302					
5	*C. gariepinus*((MG988400)	0.339	5.005	4.400	4.635				
6	*C. gariepinus* (Indonesia)	2.837	4.987	4.638	6.238	2.594			
7	*C. gariepinus* (Nigeria)	4.055	4.636	5.095	3.147	4.025	2.315		
8	Suspectred hybrid (MG988401))	0.311	4.706	4.297	4.763	0.295	2.631	4.629	

## Data Availability

The gene sequences generated in this study, is deposited to NCBI nucleotide sequences database (https://www.ncbi.nlm.nih.gov/), which can be publically accessed by using the accession number given at Table 1.

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
