# Peer review of "Invasion of African Clarias gariepinus Drives Genetic Erosion of the Indigenous C. batrachus in Bangladesh"

_biology, 2022, doi:10.3390/biology11020252_

Round 1

Reviewer 1 Report

Review Biology-1470842

The manuscript by Parvez et al. investigates on the potential hybridation of native population of C. batrachus with C. gariepinus in Bangladesh . To do that, they took muscle samples (n=20 for each group) from local market fish, classified as C. batrachus (group 1), potential hybrid based on morphological assessment: C. batrachus (group 2). Then, they compared Cytochrome C Oxidase subunit (COI) and cytochrome b (Cytb) genes between from sequenced and retrieved sequences from the species from different countries. They found that (1) the morphologically dissimilar C. batrachus showed the least genetic distance from C. gariepinus, and (2) the suspected hybrid formed sister taxa with the exotic C. gariepinus for both genes investigated. So conclude about the potential hybridation of the two species in Bangladesh.

Overall the topic is interesting and fits well the scope of the special issue. I however have some concerns regarding the manuscript. First, the acquisition of the 20 sample is not enough clear and questionable (the sample size seems quite low for such study). In some tables, the sequences deposited on the NCBI site are not clearly assigned to group (1) or (2) in the MS, so it could be difficult to follow the manuscript for the reader. Maybe I missed, but I did not understand where the authors took the sequence for C. gariepinus that they deposited on NCBI in the M&M.  In addition, the trees displayed by the authors both for COI and Cytb appeared to be poorly supported by bootstrap values (no mention of bootstrap in the manuscript). The authors should better acknowledge in their manuscript (e.g. in discussion) the limits of the study, involving higher sampling size, other genes, etc.

Specific comments are provided below.

Specific comment :

Simple abstract :

Ln 17: shares or share ?

Ln 20: predates… the “s” is only for the third person of singular, please check it in the whole MS.

Ln 25-29: not sure is as simple for wide audience.

Ln 30: higher gene ?

Graphical abstract:

The graphical abstract is lacking of clear conclusion. The reader see the tree but did not understand the main finding of the study.

Introduction:

Ln 69: half of the sea water ? What’s mean ? the salinity of sea is changing around the world.

Ln 80: At which date Clarias gasriepinus was banned ? please add the information here.

Ln 81: Is this sentence needed here ?

Ln 103: deleted “was used”.

Material and methods:

Ln 116: one sample per fish ? It needs here more details. what kind of sample from fish ? Randomly ? Please explain.

Ln 117: what preliminary fishes means ?

Ln 120: I think the authors needs to better acknowledge that (2) is referring to suspected hybrid. The end of the sentence is not very clear. No C. gaspierinus found in the fish sample ?

Ln 118-126: In my opinion, the paper will benefit of a figure to clearly show how the classification into different groups.

Ln 128: n=20 for c. batrachus population (1), and n=20 for c. batrachus population (2), what going on with the other samples ? no classification for the rest of fish ? Or they did random sampling of n=20 for each group based on 120 fish sample available ? Please clarify this point.

Ln 130: it needs more details related to method used by the authors to extract DNA. The authors could refer to previous published material.

Figure 1: I suggest to draw the figure 1a alone since it is difficult to see the morphological differences highlighted by the authors in a small panel like that (see previous comment).

Ln 154: please add purification kit number.

Table 1: where the sequences of C.gaspierinus from Bangladesh submitted by the authors are coming from ? please add in the table where sequence is related to C. batrachus (1) and (2). In the table 1, it is not very clear what sequence are referring to COI or cyt b. A solid line for separation could benefit to the understanding of the table.

Results:

3.1. what about cytb ?

Tables 2: could be better for the reader understanding that the authors disentangle groups (1) and (2) in the table.

Ln 202: only 2 numbers after the coma would be enough in the text.

Ln 211: Is not material and method ?

Table4: “The estimates of the disparity index per site  are presented for each sequence pair below the diagonal.” What about values above ?

Ln 230: add “(Indonesia)”.

Figure 2: please write maximum likelihood instead of ML in the caption.

For 3.5 and 3.6, what about bootstrap values supporting the clades ? No mention of the authors. In addition, some bootstrap values are relatively low.

The subheading 3.6 is not well displayed.

In this section, the authors state that the tree displayed in Figure 3 validated the results of tree displayed in figure 2 related to COI nucleotide but they omitted the remaining part of the tree, where batrachus (1) is close to other C. paraspinous from France and Nigeria. Please explain

Discussion:

Ln 310: potential hybriditation. Indeed potential hybridation. The authors are stating that the results displayed in the tree are weak.

Author contributions:

Please correct the contribution by deleting X.X when needed.

Author Response

Date: 09 December 2021
Manuscript number: biology-1470842
Manuscript title: Invasion of African Clarias gariepinus drives genetic erosion of the indigenous C. batrachus in Bangladesh
To respected reviewer 1
Dear Sir, 
Thanks for your constructive comments and critical review to improve our manuscript. We have tried to address all of your comments in a point-to-point manner.
Your suggestion in the methodology and the sample size is highly appreciable, we do agree with you about our sample size limitation. We have addressed this issue in the conclusion. In our future studies, we will be aware of including more samples and more genes to analyze. In the revised manuscript, new insertions/edits are highlighted in green color.  Please see the attached file
We look forward to hearing from you.
Sincerely yours,
Dr. Siriporn Pradit , The corresponding author. 

Reviewer 2 Report

Review

Paper title: Invasion of African Clarias gariepinus drives genetic erosion of the indigenous C. batrachus in Bangladesh.

The hybridization of native species with their escaped conspecifics or other close species is an important problem worldwide. The authors studied some aspects of this problem on the example of African catfish. The authors analyzed nucleotide frequencies, sequence variation, genetic divergence, transition-transversion bias and phylogenetic relationships among catfish species in Bangladesh and confirmed the occurrence of crossbreeding between an invasive species Clarias gariepinus and a native species Clarias batrachus. These findings may have important implications for the management of aquatic invasions and aquaculture in the region.

All these reasons explain the relevance of the paper by Imran Parvez and co-authors submitted to "Biology".

General scores.

The data presented by the authors are original and significant. The study is correctly designed and the authors used appropriate methods. In general, the statistical analyses are performed with good technical standards. The authors conducted careful work that may attract the attention of a wide range of specialists focused on biological invasions, ecosystem services and aquaculture management.

Specific comments.

L 18. Change “shares” to “share”

L 23. Change “government” to “the government”

L 36. Change “native population” to “the native population”

L 67. Change “environment” to “environments”

L 69. Delete “(Pradit et al., 2021)”

L 70. Change “in Indian” to “in the Indian”

L 71. Change “in Taiwan region” to “in the Taiwan region”

L 77. Change “detrimental for” to “detrimental to”

L 91. Change “Mekong Basin” to “the Mekong Basin”

L 106. Change “The COI nucleotide sequences” to “The COI nucleotide sequences are”

L 109. Change “has found effective” to “has been found effective”

L 124. Change “outer margin” to “the outer margin”

L 137. Change “was amplified” to “were amplified”

L 154. Change “manufacturer’s instructions” to “the manufacturer’s instructions”

L 159. Change “sequences of” to “sequences”

L 161. Change “In  the  NCBI” to “On  the  NCBI”

L 162. Change “Basic Local” to “The Basic Local”

L 175. Change “null hypothesis” to “a null hypothesis”

L 179. Change “identified as” to “identified”

L 184. Change “COI  gene” to “the COI  gene”

L 186. Change “COI  gene” to “the COI  gene”

L 189. Change “Philiphine” to “the Philippines”. Make this correction throughout the text and in Tables.

Table 2. Change “C. batrachus MG988401)” to “C. batrachus (MG988401)”

L 213. Change “is presented” to “are presented”

L 220. Change “COI  gene” to “the COI  gene”

L 239. Change “inferend” to “inferred”

L 248-252. The authors should separate the head of sub-section 3.6 from the text.

L 261. Change “count” to “the count”

L 276. Change “in  two species” to “in  the two species”

L 302. Change “Ratio” to “The ratio”

L 306. Change “bias 0.94” to “bias of 0.94”

L 317. Change “Thirty-species of Family” to “Thirty species of the Family”

L 323. Change “COI  gene” to “the COI  gene”

L 335. Change “human” to “a human”

L 385. “Clarias gariepinus” should be italicized.

L 426. Missing Publisher information

L 442. Missing Journal information

Author Response

Date: 09 December 2021

Manuscript number: biology-1470842

Manuscript title: Invasion of African Clarias gariepinus drives genetic erosion of the indigenous C. batrachus in Bangladesh

To

Respected reviewer 2

Thank you for your comments on the importance of this manuscript. We are glad to hear from you. We have considered your comments and revised them accordingly.

In the revised manuscript, new insertions/edits are highlighted in green color.  

We look forward to hearing from you. Please see the attached file.

Sincerely yours,

Dr. Siriporn Pradit , The corresponding author.

Round 2

Reviewer 1 Report

Comments attached

Author Response

Dear Sir, Thank you very much for your nice review, we have tried to respond to all of your specific comments. We are submitting the response letter as well as the revised manuscript. The round 2 replies were done with red color text. The green color text indicates the response of round 1. We strongly believe your comments and suggestions improved our manuscript. Thanks again

Siriporn Pradit

Round 3

Reviewer 1 Report

The authors answered my comments in this second round of revision. I still believe that the resolution of the figures is too low for publication, please enhance the low resolution of the figures. Also, the paper would benefit of english check from a native english speaker. I finally suggested to the editor to contact an expert in genetic to provide another review report.

Author Response

Dear Reviewer,

Thank you very much that you are happy with our reply of your all the questions, According to your suggestion, we have tried to improve the resolution of graphical abstracts, Figure-1 and Figure-2. The English language of the manuscript also has changed by a Native English Speaker are shown by track changed option. The speaker mostly changes little grammatical errors, very minor sentences are restructured. Please see the attached file.

Thank you very much for your nice and valuable comments to improve the manuscript. We all the authors appreciate your great review.

Sincerely

Pradit, the corresponding author
